# GRADIENT INVERSION ATTACKS BEYOND SGD

## ABSTRACT

Gradient Inversion Attack (GIA) poses a significant threat to federated learning, enabling adversaries to reconstruct private training data from the information shared during training. Prior research has predominantly focused on the vanilla SGD, where the server or an eavesdropper can directly observe true gradients. In practical deployments, however, models may be trained with adaptive optimizers (e.g., Adam, RMSProp, and AdaGrad), for which the observable signal is not raw gradients but momentum-based parameter updates. This setting remains underexplored and undermines traditional gradient-matching strategies, which struggle to recover labels and images from non-gradient updates. To address this gap, this paper explores attacks tailored to modern adaptive optimizers. We present an analytical rule for recovering labels from optimizer updates and propose an update-matching objective that optimizes dummy inputs to reproduce the observed updates. The proposed approach is general and can be directly applied to various optimizers such as Adam, AdaGrad, and RMSProp. Furthermore, we find that, despite being introduced for adaptive optimizers, the proposed objective function also yields stronger attacks in the standard SGD setting. Experiments on datasets such as ImageNet and PACS highlight the effectiveness of our method over existing gradient matching techniques.

## 1 INTRODUCTION

Gradient inversion attacks (GIAs) (Zhu et al., 2019; Geiping et al., 2020; Huang et al., 2021) have emerged as a severe threat to the privacy of federated learning (FL) (McMahan et al., 2017; Kairouz et al., 2021; Konečnỳ, 2016). These attacks allow a curious participant, such as the FL server, to reconstruct clients' private data from the optimization signals shared during model training. Specifically, in each FL round, clients train the model locally on their private data and transmit updates (e.g., gradients) to a central server (Huang et al., 2021). An adversary with access to these signals can reconstruct clients' private data by generating inputs that best match the shared updates.

GIA originates from Deep Leakage from Gradients (DLG) (Zhu et al., 2019), which formulates an optimization procedure that reconstructs an input–label pair by matching gradients computed from dummy data to those uploaded by a victim client. Subsequent works further enhanced GIAs, achieving high-fidelity recovery of batch-level inputs by introducing improved gradient matching loss (Geiping et al., 2020), analytical label restoration methods (Zhao et al., 2020; Yin et al., 2021; Ma et al., 2023; Ye et al., 2024), additional regularization terms (Geiping et al., 2020; Yin et al., 2021; Fang et al., 2023), and Generative adversarial network (GAN) priors (Jeon et al., 2021; Li et al., 2022; Fang et al., 2023; Goodfellow et al., 2020). Despite these advances, most GIA methods hinge on a key assumption: the attacker has direct access to gradients (Zhu et al., 2019; Geiping et al., 2020; Huang et al., 2021), as illustrated in the left part of Figure 1. In practice, however, FL clients may share optimizer-produced parameter updates rather than raw gradients, so this assumption holds primarily when the local optimizer is standard SGD.

Currently, adaptive optimizers like Adam (Kingma, 2014) are widely used across modern architectures (Kunstner et al., 2023), and recent studies have also incorporated adaptive optimizers into FL (Karimireddy et al., 2020). In this regime, the server typically observes optimizer-produced parameter updates rather than raw gradients, which undermines traditional GIA methods that rely on direct gradient access. Specifically, prior works typically optimize dummy inputs by minimizing a distance between dummy and true gradients (Zhu et al., 2019; Geiping et al., 2020; Huang et al., 2021) and leverage the mathematical properties of gradients to infer labels (Zhao et al., 2020; Yin

et al., 2021; Ma et al., 2023; Ye et al., 2024). When raw gradients are not exposed, these objectives are undefined, and the associated label derivations no longer apply. Consequently, even without explicit privacy design, adaptive optimizers can substantially reduce the effectiveness of current GIA techniques.

In this paper, we propose a GIA method tailored to adaptive local optimizers, as shown in the right part of Figure 1. Our approach analytically extracts labels from model updates and generates inputs supervised by the true model updates through an optimization process (illustrated in Figure 2). Specifically, we derive element-wise equations for the final-layer gradients based on the adaptive optimizer's update rule. Because the per-element solutions can be non-unique, we treat them as candidate gradient values and recover a globally consistent gradient by enforcing constraints derived from the mathematical properties of final-layer gradients. Labels are then revealed from the recovered gradients. With these recovered labels, we initialize images from random noise and optimize them to minimize the distance between the model updates induced by the generated data and the observed updates from real data, thereby reconstructing private examples. Notably, although developed for adaptive optimizers, the proposed objective function with update-matching also improves reconstruction quality in the standard SGD setting when used in place of conventional gradient matching for an initialized FL model.

To evaluate the effectiveness of our method, we conduct experiments on the ImageNet (Deng et al., 2009) and PACS (Li et al., 2017) datasets, comparing our approach with IG (Geiping et al., 2020), GIAS (Jeon et al., 2021), GIFD (Fang et al., 2023), and HFGI (Ye et al., 2024) methods. In the adaptive-optimizer (update-only) setting, our method reconstructs images that closely resemble the originals, whereas the baselines produce hardly recognizable outputs. Quantitatively, our approach improves peak signal-to-noise ratio (PSNR) by up to 7 dB over these baselines. We also observe that the method generalizes effectively across different optimizers such as Adam, RMSProp (Tieleman & Hinton, 2012), and AdaGrad (Duchi et al., 2011). Furthermore, even under standard SGD, where gradients are directly accessible, our approach outperforms the baselines by roughly 3.0 dB in PSNR.

**Contributions:** To the best of our knowledge, we are the first to show that GIAs can extend beyond FL algorithms that rely on local SGD. We propose the first GIA approach that generalizes effectively across various modern adaptive optimizers through an image optimization process and an analytical label-inference method. Additionally, the proposed update-matching objective also serves as a loss function that enables a stronger attack in the standard SGD setting.

## 2 RELATED WORK

### 2.1 GRADIENT INVERSION ATTACKS

Existing GIAs can generally be categorized into optimization-based and analysis-based methods.

**Optimization-based GIA.** Zhu et al. (2019) introduced the first optimization-based GIA, which iteratively optimizes a dummy input-label pair to minimize the distance between gradients from the true data and the dummy data. Zhao et al. (2020) enhanced this approach by analytically inferring the true labels in advance, optimizing only the input data thereafter. Geiping et al. (2020) advanced high-resolution image reconstruction on deep neural networks through a novel gradient-matching loss and a regularization term. Yin et al. (2021) achieved high-fidelity batch data recovery using batch normalization (BN) statistics. Jeon et al. (2021) improved reconstruction quality by incorporating generative adversarial network (GAN) priors. Hatamizadeh et al. (2022) successfully reconstructed data from vision transformer gradients. Li et al. (2022) showed that private information can still leak from degraded gradients under certain defense settings. Xu et al. (2022) enable efficient reconstruction across multiple epochs via an approximate GIA method. Usynin et al. (2023) improved reconstruction fidelity by introducing adversarial priors derived from attacker-controlled data. Zhu et al. (2023) proposed a surrogate model method that achieves efficient data reconstruction from weight updates accumulated over many local iterations. Fang et al. (2023) introduced GIFD, which leverages the feature domain of GAN models to enhance reconstruction quality. More recently, Ye et al. (2024) proposed a stepwise gradient matching method and a novel label restoration technique, achieving high-quality data reconstruction when duplicate labels exist within a batch.

**Analysis-based GIA.** Analysis-based GIA represents the other research direction. These attacks reconstruct training data by formulating and solving equations that link inputs with gradients (Aono et al., 2017; Zhu & Blaschko, 2020; Fowl et al., 2021; Du et al., 2024). Without the need to optimize generated data, these methods are generally more computationally efficient. However, they are typically unsuitable for batch training or may require a malicious server (Fowl et al., 2021). Additionally, they are often limited to specific network architectures (Ye et al., 2024; Fang et al., 2023). Despite their impressive performance, existing GIA methods have not considered the case where clients may use adaptive optimizers.

## 2.2 Adaptive local optimizer in Federated Learning

Adaptive optimization methods in FL can generally be classified into two categories: server-adaptive and client-adaptive methods (Jin et al., 2022). As server-adaptive methods do not change the threat model of existing GIAs, we focus on client-adaptive methods. In the literature, Reddi et al. (2020) proposed the first FL framework allowing both clients and the server to use adaptive optimizers. Liu et al. (2020) introduced momentum gradient descent as a local optimizer, requiring clients to send their optimizer states to the server for aggregation after each training round. Karimireddy et al. (2020) presented MIME, which incorporates adaptive optimizers (e.g., Adam) in local client training with shared optimizer states (e.g., momentum) across clients. Wang et al. (2021) suggested restarting local optimizer states at the start of each global training round. Sun et al. (2023) introduced client-level momentum to accelerate training and reduce heterogeneous overfitting. Wu et al. (2023) demonstrated that independently combining adaptive optimizers and local training on each client can cause divergence and proposed a momentum-based variance reduction method for local adaptive optimization. Additionally, Lewis et al. (2024) proposed a privacy-preserving approach using private adaptive optimizers, concealing the initial state of the adaptive local optimizer from the server through secure aggregation (Bonawitz et al., 2016). However, secure aggregation techniques can impose substantial communication overhead (Lewis et al., 2024; Ye et al., 2024), and an attacker may even evade the secure aggregation protocol (Pasquini et al., 2022).

## 3 Method

In this section, we first introduce the problem formulation and the threat model, by which we explain the main challenges in previous paradigms. We then present our method, which enables data reconstruction beyond the SGD setup, incorporating image optimization and label recovery.

**Problem Formulation.** Consider a model $f_\theta$ with parameters $\theta$ for image classification tasks with cross-entropy loss $\ell(\cdot)$, and a batch of training data pairs $(x, y) = \bigcup_{i=1}^{B} \{(x_i, y_i)\}$ with image $x_i$ and label $y_i$, the gradients can be calculated as $\nabla\theta = \frac{1}{B}\sum_{i=1}^{B}\nabla_\theta\ell(f_\theta(x_i), y_i)$. In a traditional SGD-based FL system, after receiving the gradients $\nabla\theta$, the attacker could invert the gradients to private data by generating tensor pair $\hat{\mathbf{x}} \in \mathbb{R}^{B \times H \times W \times C}$ and $\hat{\mathbf{y}} \in \{0,1\}^{B \times L}$ ($B, H, W, C, L$ being batch size, height, width, number of channels and class number respectively) with the following objective function:

$$\hat{\mathbf{x}}^*, \hat{\mathbf{y}}^* = \arg\min_{\hat{\mathbf{x}}, \hat{\mathbf{y}}} \mathcal{D}\left(\nabla\hat{\theta}, \nabla\theta\right) + \alpha\mathcal{P}(\hat{\mathbf{x}}), \tag{1}$$

where $\nabla\hat{\theta} = \frac{1}{B}\sum_{i=1}^{B}\nabla\ell(f_\theta(\hat{x}_i), \hat{y}_i)$ is the average gradients across the generated data, and $(\hat{x}, \hat{y}) = \bigcup_{i=1}^{B}\{(\hat{x}_i, \hat{y}_i)\}$. $\mathcal{D}(\cdot)$ is the measurement of distance. $\mathcal{P}(\hat{\mathbf{x}})$ is the image regularization term (e.g., total variation (Geiping et al., 2020)), and $\alpha$ is the corresponding coefficient. However, when FL clients leverage adaptive optimizers for local training, model updates are computed as $\theta \leftarrow \theta - \eta\mathcal{U}(\nabla\theta, s)$, where $\eta$ is the learning rate, $\mathcal{U}(\cdot)$ is the computation rule of the adaptive optimizer and $s$ is the initial state of the optimizer (e.g., momentum). Consequently, the raw gradients are no longer accessible to attackers, thereby impeding the data reconstruction methods used in previous studies.

**Threat Model.** We investigate the scenario in which an honest but curious server attempts to reveal clients' private data from their uploaded updates. The server can receive model updates from any client. Since existing adaptive FL algorithms typically unify the optimizer state across clients at the

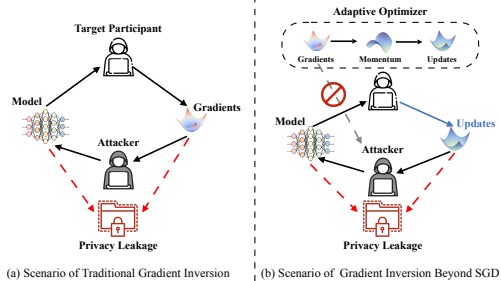

(a) Scenario of Traditional Gradient Inversion | (b) Scenario of Gradient Inversion Beyond SGD

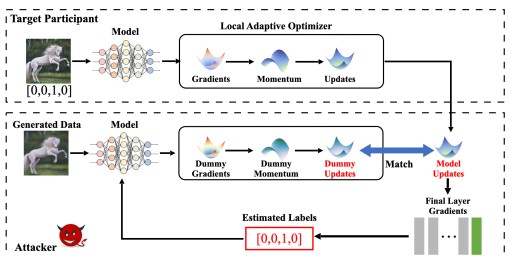

Figure 1: Attack scenarios of traditional gradient inversion attacks and gradient inversion in settings beyond SGD. In traditional SGD-based scenarios, attackers have direct access to gradients. Beyond SGD, only model updates are available to attackers, rendering **raw gradients inaccessible**.

Figure 2: Overview of our proposed method. The generated images are optimized to minimize the distance between the dummy updates and the shared updates from the target participant, with labels analytically inferred. To reveal label information, the attacker first recovers the final-layer gradients from the model updates, then extracts the labels from the recovered gradients.

beginning of each local training round (Karimireddy et al., 2020; Wu et al., 2023; Sun et al., 2023), we assume the adversary has access to the initial state (e.g., momentum) of each local optimizer but not to the state after training. For instance, the server can access the initial values of the first- and second-order momentum in the Adam optimizer, but not their values after training.

**Challenges in Previous Paradigms.** The main challenges brought by the absence of raw gradients stem from two aspects: **(1) Objective Function.** Previous works optimize generated images guided by the distance between the generated gradients and true gradients, as shown in Equation 1. However, the absence of raw gradient information prevents the calculation of these objective functions. **(2) Label Restoration.** Prior methods typically exploit the mathematical properties of gradients to derive label information. For a simple illustration, when the training data is a single image, the ground truth label can be accurately inferred through (Zhao et al., 2020):

$$c = i, \quad \text{s.t.} \quad \nabla \mathbf{W}_{\text{FC}}^{i}{}^{\top} \cdot \nabla \mathbf{W}_{\text{FC}}^{j} \leq 0, \quad \forall j \neq i, \tag{2}$$

where $\nabla \mathbf{W}_{\text{FC}}^{i}$ is the final layer gradient vector w.r.t to the weights connected to the $i_{th}$ logit. However, without access to raw gradients, these types of derivations become infeasible, hindering accurate label inference of previous methods.

To address these challenges, we iteratively optimize the generated images to search for the set of inputs that best match the true model updates, with one-hot labels analytically derived solely from the model updates. An overview of our method is shown in Figure 2.

### 3.1 IMAGE OPTIMIZATION

We reconstruct the images by iterative optimization using the following objective function:

$$\hat{\mathbf{x}}^* = \arg\min_{\hat{\mathbf{x}}} \left( 1 - \frac{\langle \mathcal{U}(\nabla\hat{\theta}, s), \mathcal{U}(\nabla\theta, s) \rangle}{\|\mathcal{U}(\nabla\hat{\theta}, s)\| \|\mathcal{U}(\nabla\theta, s)\|} + \alpha \mathcal{P}(\hat{\mathbf{x}}) \right). \tag{3}$$

Since Adam is one of the most widely used adaptive optimizers, we use it as an example for the following analysis, which can be easily generalized to other adaptive optimizers. The computation rule of Adam can be denoted as:

$$\mathcal{U}(\nabla\theta, s) = \hat{m}_t(\nabla\theta) / \left( \sqrt{\hat{v}_t(\nabla\theta)} + \epsilon \right), \tag{4}$$

where $\hat{m}_t(\nabla\theta) = \frac{\beta_1}{1-\beta_1^t} m_{t-1} + \frac{1-\beta_1}{1-\beta_1^t} \nabla\theta$, and $\hat{v}_t(\nabla\theta) = \frac{\beta_2}{1-\beta_2^t} v_{t-1} + \frac{1-\beta_2}{1-\beta_2^t} \nabla\theta^2$. $m_t$ and $v_t$ represent the first-order and second-order momentum; $\beta_1$ and $\beta_2$ are the averaging factors; and $\epsilon$ is the numerical stability term. Specifically, we start with randomly initialized dummy inputs to compute dummy gradients. Using the initial state (i.e., initial momentum values) of the local optimizer and these

dummy gradients, we calculate the dummy momentum, enabling us to derive the dummy updates. The loss is then computed by measuring the negative cosine similarity between the dummy updates and the true model updates. We incorporate total variation (Geiping et al., 2020) as the regularization term. Additionally, we apply random drop (Ye et al., 2024) to the model updates for each layer at each optimization step that requires matching.

Given the objective function, since labels cannot be reliably extracted using existing methods, an intuitive approach is to jointly optimize both images and labels using this objective function. To evaluate the feasibility of this approach, we conduct attack experiments on ImageNet with a randomly initialized ResNet-18 (He et al., 2016) model. The reconstruction results of this approach are shown in the third row in Figure 3, demonstrating that relying solely on the objective for both image and label optimization struggles to produce high-fidelity images. We attribute this to the shift of labels during optimization, a phenomenon similar to what has been observed in the prior work (Ma et al., 2023) on SGD-based cases. Therefore, we analytically recover the labels to further enhance the attack. As shown in the second row of Figure 3, our method achieves high-quality reconstructions with analytically recovered labels.

Additionally, we compare the reconstructions from our attack in the Adam setting and those from the previous method IG (Geiping et al., 2020) in the traditional SGD setup (fourth row in Figure 3). The results demonstrate that our method produces even higher-fidelity images than those obtained by the previous approach using gradient matching. In this case, since both the labels recovered by our method and the previous method are accurate, the superiority of our method stems from our new objective function in Equation 3. Notably, although FL clients do not compute or upload adaptive updates in traditional SGD settings, an attacker can actively construct adaptive updates to utilize the loss function in Equation 3. For instance, an attacker can estimate an optimizer state $s$ using historical gradient information it received. Given this state, upon receiving new gradients $\nabla\theta$,

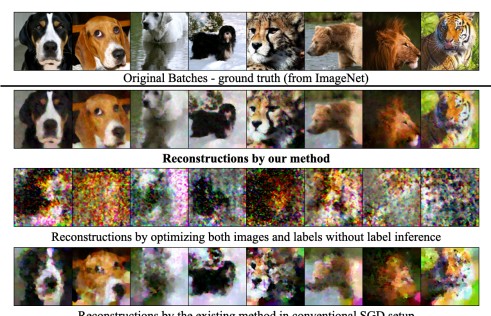

Figure 3: Reconstruction results of (1) our method with analytical label recovery, (2) joint optimization without label inference, and (3) the method from Geiping et al. (2020).

the attacker can compute $\mathcal{U}(\nabla\theta, s)$, thereby computing Equation 3. Consequently, our work not only extends gradient inversion risks to adaptive optimizers but also can serve as a stronger attack in a traditional SGD setting, which will be shown in Section 4.1 and Section 4.3, respectively.

## 3.2 Label Recovery

Previous works have demonstrated that ground-truth labels can be accurately inferred from the final-layer gradients (Zhao et al., 2020; Yin et al., 2021; Ma et al., 2023; Ye et al., 2024). Inspired by this, we recover the gradients of the final layer from the model updates, and then extract the labels from the recovered gradients. Based on Equation 4, we can formulate an equation:

$$\mathcal{U}(\nabla\theta, s) = \frac{\frac{\beta_1}{1-\beta_1^t}m_{t-1} + \frac{1-\beta_1}{1-\beta_1^t}\nabla\theta}{\sqrt{\frac{\beta_2}{1-\beta_2^t}v_{t-1} + \frac{1-\beta_2}{1-\beta_2^t}\nabla\theta^2} + \epsilon}. \tag{5}$$

Since $\epsilon$ is very small, it can typically be ignored. Additionally, attackers can legally obtain $\mathcal{U}(\nabla\theta, s)$, $m_{t-1}$, $v_{t-1}$, $\beta_1$, and $\beta_2$. Therefore, each element of $\nabla\theta$ in Equation 5 can be solved as a quadratic equation, yielding one or two solutions. Then the key is to establish sufficient constraints to identify the true gradients from these potential values.

Since solving Equation 5 yields a unique solution for some gradient elements, these can be fixed directly without additional constraints. To investigate this, we empirically tracked the number of elements with two candidate solutions over training steps. We observe that this count drops rapidly as training progresses. That is, the number of uniquely solvable elements rises quickly. This suggests that, outside the initial phase, only a small fraction of elements require extra constraints to be

determined after solving Equation 5, which simplifies gradient recovery. Guided by this observation, we first focus on the post-initial phase and then move on to the initial phase. Detailed experimental results are provided in the Appendix.

**Phase After Initial Training Stage.** In this case, we observe that elements with two candidate solutions tend to cluster in specific rows, with indices corresponding to labels present in the training batch. Guided by this observation, we establish constraints from the perspective of columns.

For a fully connected layer, we have $o = Wz + b$, where $o \in \mathbb{R}^{q \times 1}$, $W \in \mathbb{R}^{q \times p}$, $z \in \mathbb{R}^{p \times 1}$, and $b \in \mathbb{R}^{q \times 1}$. Here, $o, W, z, b, p, q$ represent the output, weights, input, bias, input dimension, and output dimension of the layer, respectively. In this context, for any row index $r \in [0, q)$, the following holds:

$$
\overline{\nabla W_r} = \frac{1}{B} \sum_{i=1}^{B} \nabla W_{i,r} = \frac{1}{B} \sum_{i=1}^{B} \frac{\partial \ell(x_i, y_i)}{\partial o_{i,r}(x_i)} z_i^T
$$

$$
\overline{\nabla b_r} = \frac{1}{B} \sum_{i=1}^{B} \nabla b_{i,r} = \frac{1}{B} \sum_{i=1}^{B} \frac{\partial \ell(x_i, y_i)}{\partial o_{i,r}(x_i)}.
$$

(6)

Based on Equation 6, the true gradients can be identified using the following constraints:

$$
\sum_r \overline{\nabla b_r} = \frac{1}{B} \sum_{i=1}^{B} \sum_r \frac{\partial l(x_i, y_i)}{\partial b_r} = 0
$$

$$
\sum_r \overline{\nabla W_{r,c}} = \frac{1}{B} \sum_{i=1}^{B} \sum_r \frac{\partial l(x_i, y_i)}{\partial W_{r,c}} = 0,
$$

(7)

where $c$ can be any column index. Specifically, we identify the true gradients of each column by selecting the unique combination of candidate values that ensures the sum of gradients in the column is zero. Next, using the recovered final-layer gradients, the labels can be extracted by leveraging existing methods (Ma et al., 2023; Ye et al., 2024).

**Initial Training Phase.** In the initial training phase, each column contains numerous elements with two candidate values. This makes it challenging to efficiently identify the true values of each gradient element using only the constraints from Equation 7. To address this, we introduce additional constraints based on the characteristics of models during the initial training phase. We first divide the training batch $(x, y)$ into $T$ subsets according to labels, specifically, $(x, y) = \{\mathbb{B}_1, \ldots, \mathbb{B}_T\}$. Next, we introduce an approximation (Ma et al., 2023) and an assumption (Yin et al., 2021).

**Approximation 1** (Inter-class Low Entanglement of Gradient Contributions). For a model in the initial training stage, the batch-averaged gradient row at index $r$ is mainly from samples of $r-th$ class in the training batch. Specifically, if $\mathbb{B}_r \neq \emptyset$, we have:

$$
\overline{\nabla b_r} = \frac{1}{B} \sum_{j=1}^{T} |\mathbb{B}_j| \overline{\nabla b_{r,\mathbb{B}_j}} \approx \frac{|\mathbb{B}_r|}{B} \overline{\nabla b_{r,\mathbb{B}_r}}
$$

$$
\overline{\nabla W_r} = \frac{1}{B} \sum_{j=1}^{T} |\mathbb{B}_j| \overline{\nabla W_{r,\mathbb{B}_j}} \approx \frac{|\mathbb{B}_r|}{B} \overline{\nabla W_{r,\mathbb{B}_r}}.
$$

(8)

**Assumption 1** (Non-negative Activation Function). *The input of the final fully connected layer is non-negative, i.e., $z_{n,i} >= 0$, where $z_{n,i}$ is the $n-th$ element of $z_i$.*

Assumption 1 holds if the preceding layer has a commonly used activation function such as ReLU (Glorot et al., 2011) or Sigmoid. Combining Approximation 1, Assumption 1, and Equation 7, for any $\overline{\nabla W_{r,c}} \neq 0$ and $\overline{\nabla b_r} \neq 0$, the gradients of the last layer satisfy the following condition:

$$
\begin{cases}
\sum_r \overline{\nabla b_r} = 0, \\
\sum_r \overline{\nabla W_{r,c}} = 0, \\
|B| \overline{\nabla b_r} >= -|\mathbb{B}_r| \\
\overline{\nabla W_{r,c}} < 0 \Rightarrow \mathbb{B}_r \neq \emptyset \\
(|B| \overline{\nabla b_r} \approx -|\mathbb{B}_r|) \wedge (\overline{\nabla b_r} < 0) \iff \mathbb{B}_r \neq \emptyset.
\end{cases}
$$

Table 1: Quantitative comparison of our method with baselines on images of the ImageNet and PACS dataset. We calculate the average value of metrics on reconstructed images. ↑: the higher the metric, the better the performance. ↓: the lower the metric, the better the performance.

| Metric | ImageNet | | | | | PACS | | | | |
|---|---|---|---|---|---|---|---|---|---|---|
| | IG | GIAS | GIFD | HFGI | **Ours** | IG | GIAS | GIFD | HFGI | **Ours** |
| MSE↓ | 0.0786 | 0.0806 | 0.0778 | 0.2030 | **0.0099** | 0.0790 | 0.1067 | 0.1069 | 0.1926 | **0.0142** |
| PSNR↑ | 11.1101 | 11.1840 | 11.3225 | 7.0102 | **20.5985** | 11.1434 | 9.9322 | 9.9342 | 7.2045 | **19.0078** |
| SSIM↑ | 0.3074 | 0.1524 | 0.1579 | 0.2251 | **0.5026** | 0.3619 | 0.1604 | 0.1700 | 0.2286 | **0.5192** |
| LPIPS (AlexNet)↓ | 0.7694 | 0.8737 | 0.8122 | 1.2071 | **0.4230** | 0.6477 | 0.8796 | 0.8308 | 1.2178 | **0.3919** |
| LPIPS (VGG)↓ | 0.7974 | 0.8044 | 0.8019 | 0.8371 | **0.5324** | 0.7736 | 0.8348 | 0.8273 | 0.8546 | **0.5202** |

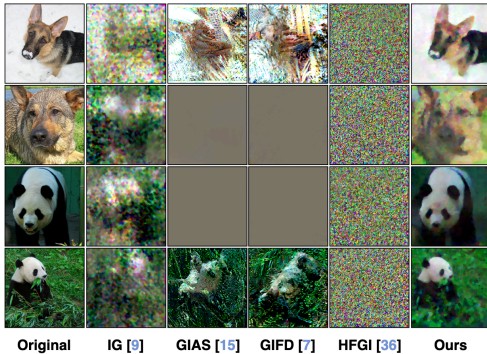 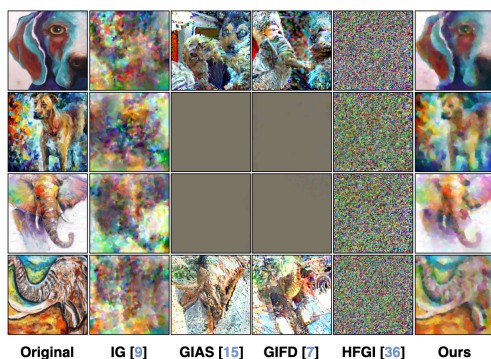

(a) ImageNet Attack with Raw Gradient Inaccessible     (b) PACS Attack with Raw Gradient Inaccessible

Figure 4: Visual comparison of our method with baselines on images of the ImageNet and PACS dataset with $B = 4$.

Based on the above properties together with the candidate gradient values, we can identify the label index $r$ where $\mathbb{B}_r \neq \emptyset$, i.e., the label indices present in the training batch and their corresponding gradients $\overline{\nabla b_r}$. For these indices, since $|B|\overline{\nabla b_r} \approx -|\mathbb{B}_r|$, we only need to calculate $-|B|\overline{\nabla b_r}$ and round the result to the nearest integer. This value corresponds to the number of data points in the batch with that label, allowing us to recover the labels for the entire training batch. The detailed proof is provided in Appendix A.

## 4 EXPERIMENTS

To evaluate the effectiveness of our method in reconstructing private data from model updates, we conduct attack experiments on image classification tasks using the ImageNet ILSVRC 2012 (Deng et al., 2009) and PACS (Li et al., 2017) datasets with 224×224-pixel images. We use ResNet-18 (He et al., 2016) as the FL model and Adam as the default local adaptive optimizer, with additional experiments conducted using other adaptive optimizers, such as RMSprop and Adagrad. We set $B = 4$ by default and provide additional experiments with different batch sizes in the Appendix.

**Implementation Details.** In our attack method, we use Adam to optimize the generated inputs. The learning rate is initialized at 0.1 with step decay. For each attack experiment, we optimize the batch for 24000 iterations, randomly dropping 30% of the model updates (Ye et al., 2024) for each layer at each step. The value of $\alpha$ in Equation 3 is set to $5 \times 10^{-3}$ for ImageNet and $10^{-2}$ for PACS. To simulate the initial state of the local optimizer, we trained the model from scratch and recorded the optimizer state at various training stages. Please refer to Appendix E for more details.

Besides (1) visual comparison, we report the following quantitative metrics to assess reconstruction quality: (2) Mean Square Error (MSE), (3) Peak Signal-to-Noise Ratio (PSNR), (4) Similarity Structural Index Measure (SSIM), and (5) Learned Perceptual Image Patch Similarity (LPIPS) (Zhang et al., 2018) computed with AlexNet (Krizhevsky et al., 2012) and VGG network (Simonyan & Zisserman, 2014).

Table 2: Quantitative results of our method and IG with different adaptive optimizers.

| Optimizer | Method | Metric | | | |
|-----------|--------|--------|--------|--------|--------|
| | | PSNR↑ | LPIPS↓ | SSIM↑ | MSE↓ |
| Momentum | IG | 10.6764 | 0.7759 | 0.2891 | 0.0881 |
| | Ours | 17.0135 | 0.5577 | 0.4128 | 0.0226 |
| Adagrad | IG | 9.6361 | 0.7839 | 0.2847 | 0.1098 |
| | Ours | 18.7233 | 0.4782 | 0.4265 | 0.0154 |
| RMSProp | IG | 11.8515 | 0.7676 | 0.3033 | 0.0667 |
| | Ours | 20.4626 | 0.4590 | 0.4993 | 0.0100 |

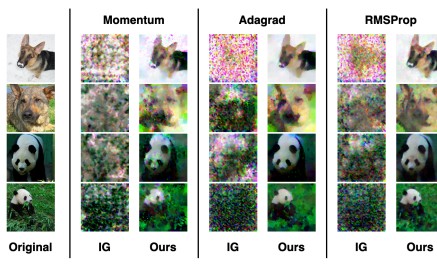

Figure 5: Reconstructions with different adaptive optimizers.

### 4.1 COMPARISON WITH GRADIENT-BASED ATTACKS

We compare our proposed method with the following state-of-the-art baselines: (1) IG by Geiping et al. (2020), (2) GIAS by Jeon et al. (2021), (3) GIFD by Fang et al. (2023), and (4) HFGI by Ye et al. (2024). Since the gradient-matching loss of the baselines cannot be calculated, we adjust their objectives from gradient matching to model-update matching. For GIAS and GIFD, we use a pre-trained BigGAN (Brock, 2018). For IG, the label inference method is inapplicable when $B > 1$. For HFGI, its label inference method causes program failure when only adaptive updates are available. Consequently, we jointly optimize the labels with the images for these two baselines. Following previous works (Zhu et al., 2019; Fang et al., 2023; Geiping et al., 2020), we set the model parameters to be randomly initialized by default. We use the optimizer state after the first step of the second training epoch to simulate the target client's optimizer state. We also conduct experiments to examine the impact of model parameters and optimizer state at different training stages, with results provided in Appendix C. Since FL clients may not transmit BN statistics computed on their private data, all experiments are conducted without the BN prior proposed by Yin et al. (2021).

We present a visual and quantitative comparison of our method with baselines in Figure 4 and Table 1. Overall, our method outperforms baseline GIAs for data reconstruction when only model updates from Adam are available. As shown in Figure 4, for both the ImageNet and PACS datasets, baseline methods struggle to generate meaningful content, while our method produces images closely resembling the real ones. In the quantitative comparison shown in Table 1, our method surpasses the best baseline results by nearly 8.0 dB, 0.15, 0.25, and 0.25 in terms of PSNR, SSIM, LPIPS (AlexNet), and LPIPS (VGG), respectively.

### 4.2 DIFFERENT ADAPTIVE OPTIMIZERS

In addition to Adam, we evaluate our method using other adaptive optimizers, including RMSprop, Adagrad, and SGD with momentum. Since SGD with momentum only applies a linear transformation to the original gradients, these gradients can be directly recovered through a simple linear transformation of the model updates, given the initial state of the optimizer. Accordingly, we match the recovered gradients for this optimizer. For all other cases, we match the model updates. The visual and quantitative results of IG and our method are shown in Figure 5 and Table 2, respectively. We find that our method generalizes well across different optimizers. As shown in Figure 5 and Table 2, IG struggles to reconstruct high-quality images, whereas our method generates images similar to the real ones across all three adaptive optimizers.

### 4.3 COMPARISON IN TRADITIONAL SGD SETUP

To further validate that the proposed objective improves reconstruction quality, we evaluate in the standard SGD setting, where raw gradients are accessible. Specifically, we provide these gradients to all baselines so that they can use their native gradient-matching objectives, whereas our method deliberately retains the update-matching objective in Equation 3 for Adam update matching.

In the standard SGD setting, our method still achieves superior performance. As shown in Table 3 and Figure 6, IG's performance varies with image content—some images are reconstructed well, whereas others degenerate into noise. GIAS and GIFD, while effective on 64×64 images Fang et al. (2023);

Table 3: Quantitative comparison of our method with baselines on images of the ImageNet and PACS dataset in the traditional SGD setting, where **with the raw gradients are directly accessible**.

| Metric | ImageNet | | | | | PACS | | | | |
|---|---|---|---|---|---|---|---|---|---|---|
| | IG | GIAS | GIFD | HFGI | **Ours** | IG | GIAS | GIFD | HFGI | **Ours** |
| MSE↓ | 0.0430 | 0.0398 | 0.0321 | 0.0086 | **0.0046** | 0.0271 | 0.0250 | 0.0373 | 0.0150 | **0.0034** |
| PSNR↑ | 14.3207 | 14.0083 | 15.0285 | 20.7997 | **23.9195** | 17.3832 | 16.3876 | 14.8280 | 19.2953 | **24.8643** |
| SSIM↑ | 0.3782 | 0.3666 | 0.3923 | 0.5211 | **0.5744** | 0.4662 | 0.4631 | 0.4249 | 0.5111 | **0.6303** |
| LPIPS (AlexNet)↓ | 0.7293 | 0.6558 | 0.6094 | 0.4826 | **0.3886** | 0.5600 | 0.4690 | 0.5126 | 0.4490 | **0.2838** |
| LPIPS (VGG)↓ | 0.7161 | 0.6746 | 0.6436 | 0.5458 | **0.4681** | 0.5938 | 0.5504 | 0.5874 | 0.5410 | **0.4049** |

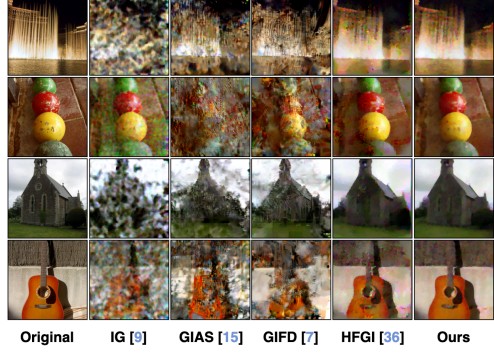 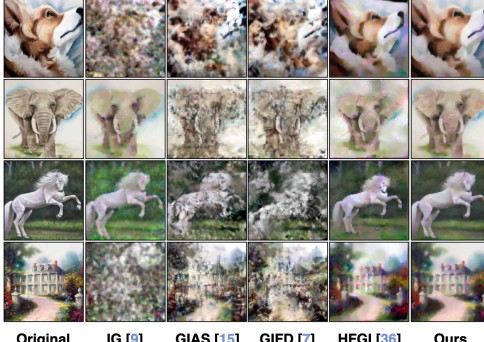

(a) ImageNet Attack with Raw Gradient Accessible  (b) PACS Attack with Raw Gradient Accessible

Figure 6: Visual comparison of our method with baselines on images of the ImageNet and PACS dataset in the traditional SGD setting, where **with the raw gradients are directly accessible**. In this case, the proposed method still outperforms existing methods.

Jeon et al. (2021), often produce blurry results at 224×224 resolution. HFGI generates visually similar images to the real ones, yet still underperforms compared to our method in both qualitative and quantitative evaluations. Quantitatively, our method outperforms the best baseline by approximately 3.0 dB in PSNR, 0.05 in SSIM, 0.09 in LPIPS (AlexNet), and 0.08 in LPIPS (VGG).

## 4.4 ABLATION STUDIES

We conduct ablations on ImageNet to assess the contribution of each component by removing label inference (w/o Label) and by replacing the proposed objective function (w/o Objective). In the w/o Label setting, labels and images are optimized jointly. In the w/o Objective setting, the images are optimized to minimize the distance between the gradients computed from the generated data and the observed model updates. The results in Table 4 show that both the proposed objective function and label-inference method are critical for high-quality reconstructions.

Table 4: Quantitative results of ablation study.

| Method | Metric | | | |
|---|---|---|---|---|
| | PSNR↑ | LPIPS↓ | SSIM↑ | MSE↓ |
| w/o Label | 11.3737 | 0.7127 | 0.3211 | 0.0763 |
| w/o Objective | 11.4221 | 0.8413 | 0.2471 | 0.0776 |
| Ours | **19.6252** | **0.4614** | **0.4981** | **0.0125** |

## 5 CONCLUSION

In this paper, we introduce an approach that can reconstruct private training data in FL from model updates produced by adaptive optimizers. This relaxes a common assumption in prior GIAs that limits attacks to SGD-based FL. The attack builds its success on an optimization-based image-generating technique alongside an analytical label-recovery method. Our experimental results on two image classification datasets demonstrate the effectiveness of our method across various adaptive optimizers and its superiority over existing attacks. We hope this work contributes to the development of more secure FL systems in the future.

## ETHICS STATEMENT

This work examines privacy risks in federated learning by extending the scope of gradient inversion attacks beyond FL algorithms that rely solely on local SGD. We also discuss corresponding mitigation strategies in the Appendix to inform defense design rather than facilitate misuse. By revealing previously underexplored attack surfaces and outlining feasible defenses, our goal is to contribute to the development of safer and more privacy-preserving federated learning systems in practice. This study involves no human subjects or personal data.

## REPRODUCIBILITY STATEMENT

The experimental settings are provided in Section 4 and the Appendix. All datasets used in this paper are publicly available. A link to an anonymous downloadable source code is provided in the Appendix.

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

## A. DETAILED PROOF

### A.1. PROOF OF EQUATION 7

For the final layer in a model with multi-class classification tasks. We have $o = Wz + b$, where $o, W, z, b$ represent the output, weights, input, and bias of this layer, respectively. For any data index $i$ and row $r$ in $W$, we have:

$$\frac{\partial \ell(x_i, y_i)}{\partial W_r} = \frac{\partial \ell(x_i, y_i)}{\partial o_{i,r}} z_i^T, \tag{9}$$

and for any row $r$ in $b$, we have:

$$\frac{\partial \ell(x_i, y_i)}{\partial b_r} = \frac{\partial \ell(x_i, y_i)}{\partial o_{i,r}}. \tag{10}$$

When the output $o_i$ from the final layer is passed through softmax and cross-entropy, the gradients of $o_{i,r}$ for any given input $x_i$ with hard label $y_i$ are:

$$\frac{\partial l(x_i, y_i)}{\partial o_{i,r}} = \begin{cases} p_{i,r} - 1 & \text{if } r = y_i \\ p_{i,r}, & \text{if } r \neq y_i \end{cases}, \tag{11}$$

where $p_i$ is the predicted probability for the $i_{th}$ class after applying softmax.
Combining Equation 9, Equation 10, and Equation 11, we obtain:

$$\frac{\partial l(x_i, y_i)}{\partial W_r} = \begin{cases} (p_{i,r} - 1) z_i^T & \text{if } r = y_i \\ p_{i,r} z_i^T, & \text{if } r \neq y_i \end{cases}$$

$$\frac{\partial l(x_i, y_i)}{\partial b_r} = \begin{cases} p_{i,r} - 1 & \text{if } r = y_i \\ p_{i,r}, & \text{if } r \neq y_i \end{cases}. \tag{12}$$

For any given data point $x_i$ and column index $c$, since $\sum_r p_{i,r} = 1$, we have:

$$\sum_r \frac{\partial l(x_i, y_i)}{\partial W_{r,c}} = 0$$

$$\sum_r \frac{\partial l(x_i, y_i)}{\partial b_r} = 0. \tag{13}$$

That is, the gradients of the bias in the last layer must sum to zero, and the sum of each column in the weight gradients of the last layer must also be zero. Since the sum of gradients for each data point is zero, averaging across all data points in a batch will also yield zero:

$$\sum_r \overline{\nabla b_r} = \frac{1}{B} \sum_{i=1}^{B} \sum_r \frac{\partial l(x_i, y_i)}{\partial b_r} = 0$$

$$\sum_r \overline{\nabla W_{r,c}} = \frac{1}{B} \sum_{i=1}^{B} \sum_r \frac{\partial l(x_i, y_i)}{\partial W_{r,c}} = 0. \tag{7}$$

Consequently, applying the constraint in Equation 7 column by column allows us to filter out incorrect candidate solutions and obtain the true gradients in the phase after the initial training stage.

### A.2 PROOF OF CONDITIONS IN THE INITIAL TRAINING PHASE

For any row index r, the average gradients of the bias across a batch of data can be denoted as:

$$\overline{\nabla b_r} = \frac{1}{B} \sum_{i=1}^{B} \nabla b_{i,r} = \frac{1}{B} \sum_{j=1}^{T} |\mathbb{B}_j| \overline{\nabla b_{r, \mathbb{B}_j}}, \tag{14}$$

where $\overline{\nabla b_{r, \mathbb{B}_j}} = \frac{1}{|\mathbb{B}_j|} \sum_{i \in \mathbb{B}_j} \nabla b_{i,r}$, (for any $\mathbb{B}_j \neq \emptyset$) is the averaged gradients across the subset $\mathbb{B}_j$. According to Equation 12, we have

$$\overline{\nabla b_{r, \mathbb{B}_j}} = \begin{cases} \frac{1}{|\mathbb{B}_j|} \sum_{i \in \mathbb{B}_j} p_{i,r} - 1 & \text{if } j = r \\ \frac{1}{|\mathbb{B}_j|} \sum_{i \in \mathbb{B}_j} p_{i,r}, & \text{if } j \neq r \end{cases}. \tag{15}$$

Since $p_{i,r} >= 0$, we obtain

$$|B|\overline{\nabla b_r} >= -|\mathbb{B}_r|. \tag{16}$$

For any row index $r$, the average gradients of the weights across a batch of data can be denoted as:

$$\overline{\nabla W_r} = \frac{1}{B} \sum_{i=1}^{B} \nabla W_{i,r} = \frac{1}{B} \sum_{j=1}^{T} |\mathbb{B}_j| \overline{\nabla W_{r,\mathbb{B}_j}}, \tag{17}$$

where $\overline{\nabla W_{r,\mathbb{B}_j}}$, is the averaged gradients across the subset $\mathbb{B}_j$. Combining Equation 12, we have

$$\overline{\nabla W_{r,\mathbb{B}_j}} = \begin{cases} \frac{1}{|\mathbb{B}_j|} \sum_{i \in \mathbb{B}_j} (p_{i,r} - 1) z_i^T & \text{if } j = r \\ \frac{1}{|\mathbb{B}_j|} \sum_{i \in \mathbb{B}_j} p_{i,r} z_i^T, & \text{if } j \neq r \end{cases}. \tag{18}$$

Based on Assumption 1, each element in $z_i$ is not negative. Additionally, $p_{i,r} >= 0$. Consequently, for any row index $r$, if $|\mathbb{B}_r| = 0$, $\overline{\nabla W_r} >= 0$. Therefore, we obtain that, for any row index $r$ and column index $c$,

$$\overline{\nabla W_{r,c}} < 0 \Rightarrow \mathbb{B}_r \neq \emptyset. \tag{19}$$

On the basis of Approximation 1, we can further simplify the Equation 14 to be:

$$\overline{\nabla b_r} = \frac{1}{B} \sum_{i=1}^{B} \nabla b_{i,r} \approx \frac{|\mathbb{B}_r|}{B} \overline{\nabla b_{r,\mathbb{B}_r}}. \tag{20}$$

Since $p_{i,r} \in [0,1]$, and in the initial training stage, $p_{i,r}$ is typically much smaller than 1, $\frac{1}{|\mathbb{B}_j|} \sum_{i \in \mathbb{B}_j} p_{i,r} - 1 < 0$. Combining Equation 20 and Equation 15, we obtain

$$\overline{\nabla b_r} < 0 \iff \mathbb{B}_r \neq \emptyset. \tag{21}$$

Additionally, as in the initial training stage, $p_{i,r} - 1 \approx -1$, for $|\mathbb{B}_r| \notin \emptyset$, we have

$$\overline{\nabla b_r} \approx \frac{|\mathbb{B}_r|}{B} \overline{\nabla b_{r,\mathbb{B}_r}} = \frac{1}{B} \left( \sum_{i \in \mathbb{B}_r} p_{i,r} - |\mathbb{B}_r| \right) \approx -\frac{|\mathbb{B}_r|}{B}. \tag{22}$$

Finally, combining Equations 7, 16, 19, 21, and 22, we obtain

$$\begin{cases} \sum_r \overline{\nabla b_r} = 0, \\ \sum_r \overline{\nabla W_{r,c}} = 0, \\ |B|\overline{\nabla b_r} >= -|\mathbb{B}_r| \\ \overline{\nabla W_{r,c}} < 0 \Rightarrow \mathbb{B}_r \neq \emptyset \\ (|B|\overline{\nabla b_r} \approx -|\mathbb{B}_r|) \wedge (\overline{\nabla b_r} < 0) \iff \mathbb{B}_r \neq \emptyset. \end{cases}$$

### A.3 SOLUTION OF EQUATION 5

Given the equation for the final-layer gradients formulated in Section 3.2 [1]:

$$\mathcal{U}(\nabla\theta, s) = \frac{\frac{\beta_1}{1-\beta_1^t} m_{t-1} + \frac{1-\beta_1}{1-\beta_1^t} \nabla\theta}{\sqrt{\frac{\beta_2}{1-\beta_2^t} v_{t-1} + \frac{1-\beta_2}{1-\beta_2^t} \nabla\theta^2} + \epsilon}, \tag{5}$$

we let

$$\alpha' = \frac{\beta_1}{1-\beta_1^t} m_{t-1}, \quad \alpha = \frac{1-\beta_1}{1-\beta_1^t},$$

$$\gamma' = \frac{\beta_2}{1-\beta_2^t} v_{t-1}, \quad \gamma = \frac{1-\beta_2}{1-\beta_2^t}, \quad U = \mathcal{U}(\nabla\theta, s).$$

Then Equation 5 (with $\epsilon \approx 0$ neglected) can be rewritten as

$$U = \frac{\alpha' + \alpha \nabla\theta}{\sqrt{\gamma' + \gamma (\nabla\theta)^2}}. \tag{23}$$

---

[1]In this subsection, $\nabla\theta$ denotes an element of the gradients, specifically $\nabla\theta_w$. For simplicity, we omit the subscript in our notation.

Rearranging and squaring both sides yields a quadratic equation:

$$(U^2\,\gamma - \alpha^2)\,\nabla\theta^2 \,-\, 2\,\alpha'\,\alpha\,\nabla\theta \,+\, \left(U^2\,\gamma' - (\alpha')^2\right) \,=\, 0.$$

Solving for $\nabla\theta$ gives

$$\nabla\theta \,=\, \frac{2\,\alpha'\,\alpha \,\pm\, \Phi}{2\left(U^2\,\gamma - \alpha^2\right)}, \tag{24}$$

where

$$\Phi \,=\, \sqrt{\left(2\,\alpha'\,\alpha\right)^2 \,-\, 4\left(U^2\,\gamma - \alpha^2\right)\left(U^2\,\gamma' - (\alpha')^2\right)}.$$

Since $\sqrt{\gamma' + \gamma(\nabla\theta)^2}$ is nonnegative, we choose the root ensuring $\alpha' + \alpha\,\nabla\theta$ shares the same sign as $U$.

## B. VISUAL RESULTS OF ABLATION STUDIES

In addition to the quantitative comparisons in Table 4, we show the visual results of our ablation studies on ImageNet in Figure 7. Consistent with the findings in Table 4, removing either the proposed objective function or the label inference method significantly degrades reconstruction quality. This indicates that both components of our method are essential for successful data reconstruction.

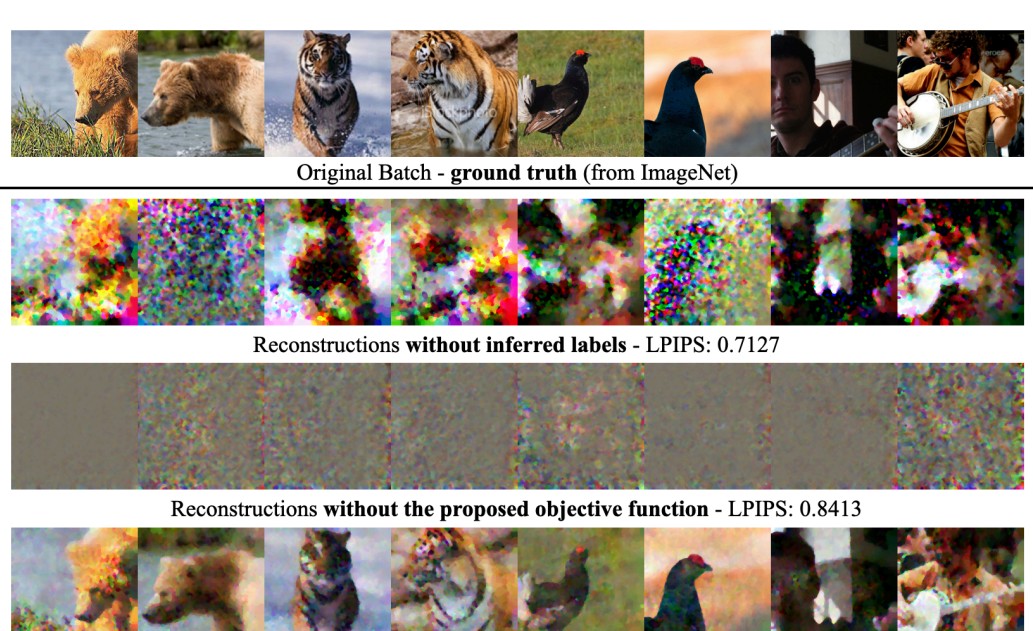

Original Batch - **ground truth** (from ImageNet)

Reconstructions **without inferred labels** - LPIPS: 0.7127

Reconstructions **without the proposed objective function** - LPIPS: 0.8413

Reconstructions - **our method** - LPIPS: 0.4614

Figure 7: Visual results of ablation study on images of ImageNet.

## C. EFFECT OF BATCH SIZE

We then conduct experiments on ImageNet to evaluate the effect of batch size on the quality of recovered images. The average LPIPS values, computed using AlexNet across images in the batches from our method and IG (Geiping et al., 2020), are presented in Figure 8. The LPIPS values increase with larger batch sizes, indicating a decline in reconstruction quality for both our method and IG. However, the reconstructions generated by our method remain similar to the original images up to $B = 16$. In contrast, the IG method achieves high-fidelity reconstructions only at $B = 1$, with reconstructions becoming unrecognizable as batch size increases.

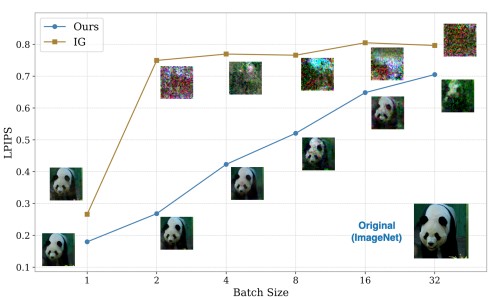

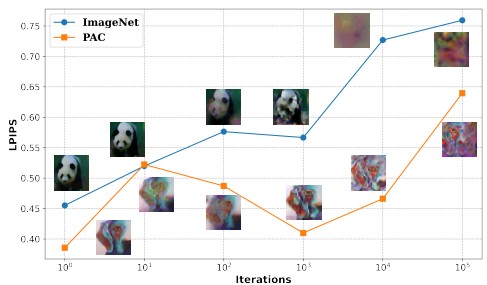

Figure 8: The impact of different batch sizes on LPIPS value.

Figure 9: The impact of different training stages on LPIPS value.

## D. EFFECT OF TRAINING STAGE ON IMAGE RECONSTRUCTIONS

For the experiments in our main paper, we used a randomly initialized model. In this section, we evaluate how different training stages affect the image reconstructions of our attack. Specifically, we train the model on ImageNet (Deng et al., 2009) from scratch and record the model parameters and optimizer state after the $1^{st}, 10^{th}, 100^{th}, 1000^{th}, 10,000^{th}$, and $100,000^{th}$ training iterations. We then conduct attack experiments using the model parameters and optimizer state from each of these stages. The batch size is set to 4, and the ground truth images are identical to those shown in Figure 4.

Figure 9 illustrates the performance of our method at different training stages, evaluated using the average LPIPS values computed with AlexNet. As training progresses, the LPIPS value generally increases, reflecting a gradual decrease in similarity between the reconstructed and real images. Specifically, the reconstructed images exhibit high similarity to the real images for the model immediately after the first iteration, similar to results obtained with an untrained model. In contrast, after 100,000 training iterations, our method reveals little information about the ground truth.

Besides, for a trained model and its corresponding optimizer state, reconstructions on the PACS dataset are significantly better than those on the ImageNet dataset. For instance, at the $10,000^{th}$ iteration, the generated data for ImageNet conveys virtually no information, whereas the generated data for PACS still retains some similarity to the real data. We attribute this to the PACS dataset having a distribution distinct from that of the ImageNet dataset used during model training. This difference likely results in larger gradient magnitudes for PACS data on the trained model, making it easier to match model updates and reveal information about the real data.

## E. EFFECT OF TRAINING STAGE ON CANDIDATE GRADIENT VALUES

We experimentally investigated the relationship between the number of elements with two candidate solutions after solving Equation 5 and the model training process. Specifically, we trained ResNet18 and ResNet34 (He et al., 2016) models from scratch on the ImageNet dataset using the Adam optimizer. At various training stages, we recorded the model parameters and optimizer states and randomly sampled batches of size 32 from the dataset to compute the corresponding gradients. As shown in Figures 10, the number of elements with only one candidate solution increases rapidly as training progresses. This observation indicates that, beyond the initial training phase, solving Equation 5 alone determines the values of most, but not all, gradient elements. The division of label inference into two cases, as discussed in our main paper, is motivated by this finding.

Notably, after the initial training stage, we experimentally observe that the remaining elements with two candidate solutions tend to cluster within specific rows, often corresponding to certain input data labels. The gradient values in these rows can be crucial for estimating the number of data samples associated with the corresponding labels. Therefore, while most elements can be determined solely by solving Equation 5, it remains necessary to resolve the values of the remaining unsolved elements for label inference.

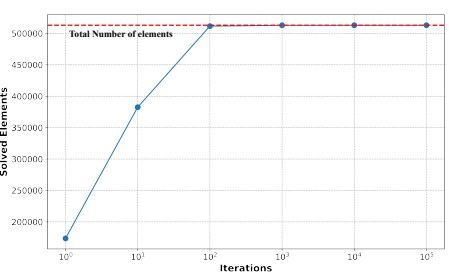

(a) ResNet-18 (Phase including the initial training stage)

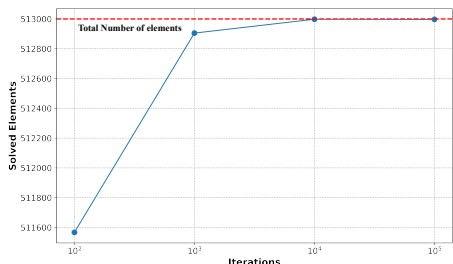

(b) ResNet-18 (Phase beyond the initial training stage)

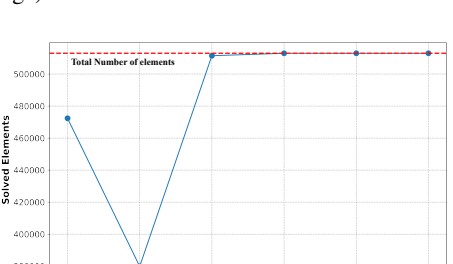

(c) ResNet-34 (Phase including the initial training stage)

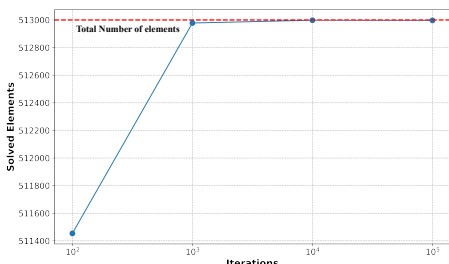

(d) ResNet-34 (Phase beyond the initial training stage)

Figure 10: Number of elements in final layer gradients can be uniquely determined by solving equation 5.

## F. ADDITIONAL IMPLEMENTATION DETAILS

For all attack experiments, we set batch size $B = 4$ by default, except in Section 4.3, where $B$ is set to 1 to maximize the effect of previous attacks. Since duplicate labels increase the difficulty of reconstruction (Ye et al., 2024), we ensure that in our Section 4 experiments. When $B \geq 4$, each batch contains labels corresponding to at least two distinct data points. Except for the experiments in Sections C and D, where the impact of the training stage is explored, FL model parameters are randomly initialized following previous works (Zhu et al., 2019; Zhao et al., 2020; Fang et al., 2023; Geiping et al., 2020; Jeon et al., 2021), and the optimizer state after the first step of the second training epoch is used to simulate the target client's optimizer state. We use Adam for image optimization, initializing the learning rate at 0.1 and decaying it at $\frac{3}{8}$, $\frac{5}{8}$ and $\frac{7}{8}$ of the total training iterations. The generated images are initialized following a Gaussian distribution and optimized based on the sign of their gradients computed using Equation 3. All experiments were conducted using NVIDIA RTX A5000, TITAN RTX, and GeForce RTX 4090 GPUs. The code is available at: https://anonymous.4open.science/r/5a0da49a4-CC07

## G. EFFECT OF RANDOM INITIALIZATION

For the experiments in our main paper, for each reconstruction, we perform four separate runs using different randomly initialized dummy images and select the result with the lowest reconstruction loss as the final output, following prior works (Zhu et al., 2019; Fang et al., 2023; Geiping et al., 2020). Since the effectiveness of gradient inversion attacks can be affected by random initializations, we evaluate our method's robustness to this factor in this section. Specifically, we further conduct five independent runs for the experiments in Section 4.1. For each run, we used a different random seed to initialize both the model parameters and the dummy images. Unlike our main experiments, each run in this setting uses only a single set of initialized dummy images without multiple trials. The evaluation results are summarized in Table 5.

As shown, compared with the setting in the main paper, where we perform four trials and select the one with the lowest reconstruction loss, the five independent single-trial runs (each with a different

Table 5: The impact of random initialization. The *Mean* column reports the average over five runs, with the values in parentheses (*(Main paper result)*) denoting the corresponding results reported in the main paper (results with the lowest reconstruction loss). *SD* and *SE* denote standard deviation and standard error, respectively.

| Metric | Mean (Main paper result) | SD | SE |
|---|---|---|---|
| LPIPS_V ($\downarrow$) | 0.5674 (0.5324) | 0.0316 | 0.0141 |
| LPIPS_A ($\downarrow$) | 0.4818 (0.4230) | 0.0517 | 0.0231 |
| PSNR ($\uparrow$) | 19.8645 (20.5985) | 0.8519 | 0.3810 |
| MSE ($\downarrow$) | 0.0115 (0.0099) | 0.0017 | 0.0008 |
| SSIM ($\uparrow$) | 0.4900 (0.5026) | 0.0168 | 0.0075 |

checkpoint and randomly initialized dummy image set) show only a slight drop in average performance. Moreover, the variability across random seeds is modest: LPIPS-V $0.5674 \pm 0.0316$, LPIPS-A $0.4818 \pm 0.0517$, PSNR $19.8645 \pm 0.8519$ dB, MSE $0.0115 \pm 0.0017$, and SSIM $0.4900 \pm 0.0168$ (mean $\pm$ SD). This indicates that different seeds yield similar outcomes, and the reconstructions consistently resemble the ground truth closely. These results demonstrate the good repeatability of our method under random initialization.

## H. RUNTIME COMPARISON

We report the runtime per reconstruction under the experimental setting described in Section 4, all measured in our experimental environment. Each runtime is measured for a single reconstruction without restart with a single GPU. As shown in Table 6, compared to IG, since our method needs to compute an adaptive optimizer update at each loss computing step, it requires more time (from 0.16h to 0.36h). However, it remains lightweight in absolute terms and is comparable to HFGI. Moreover, compared with approaches that introduce an additional generative model (GIFD and GIAS), our method is more lightweight, requiring less running time.

Table 6: Runtime per reconstruction.

| | IG | GIAS | GIFD | HFGI | Ours |
|---|---|---|---|---|---|
| GPU Hour (h) | 0.16 | 0.97 | 0.64 | 0.37 | 0.36 |

## I. EFFECT OF LABEL DUPLICATION

Prior work has shown that duplicate labels increase the difficulty of gradient-based reconstruction (Ye et al., 2024). We therefore examine how the degree of label duplication within a mini-batch affects our method. In the main paper setting, two out of four images share the same label (pairwise duplication). Here, we vary the duplication level (e.g., all labels distinct, two pairs, and all labels identical) and report the corresponding reconstruction performance in Table 7. As shown, greater label duplication leads to a slight degradation in reconstruction quality. Nevertheless, even in the extreme case where all four samples share the same label, our method still achieves a reasonably high similarity to the original data. Specifically, the reconstructed images retain a PSNR of 19.80 dB.

Table 7: Effect of label duplication on reconstruction quality. The numbers in the 1st column indicate within-batch label multiplicities; e.g., 3+1 means three samples share the same label and one differs.

| Label Duplication | LPIPS-V $\downarrow$ | LPIPS-A $\downarrow$ | PSNR (dB) $\uparrow$ | MSE $\downarrow$ | SSIM $\uparrow$ |
|---|---|---|---|---|---|
| All-same | 0.6222 | 0.5701 | 19.80 | 0.0111 | 0.4929 |
| 3+1 | 0.5542 | 0.4800 | 20.91 | 0.0090 | 0.5203 |
| 2+2 | 0.5324 | 0.4230 | 20.60 | 0.0099 | 0.5026 |
| All-different | 0.4928 | 0.3949 | 23.90 | 0.0042 | 0.5798 |

## J. MORE RECONSTRUCTION RESULTS

We show additional reconstruction results on images from ImageNet (Deng et al., 2009), PACS (Li et al., 2017), and Web (Ye et al., 2024) in Figure 11. For PACS, the reconstruction results for images in the art painting style are reported in our main paper. In this section, we extend our experiments to include photo and cartoon-style images. We can find that the reconstructed images remain similar to these additional ground truth images, further demonstrating the effectiveness of our methods across various datasets.

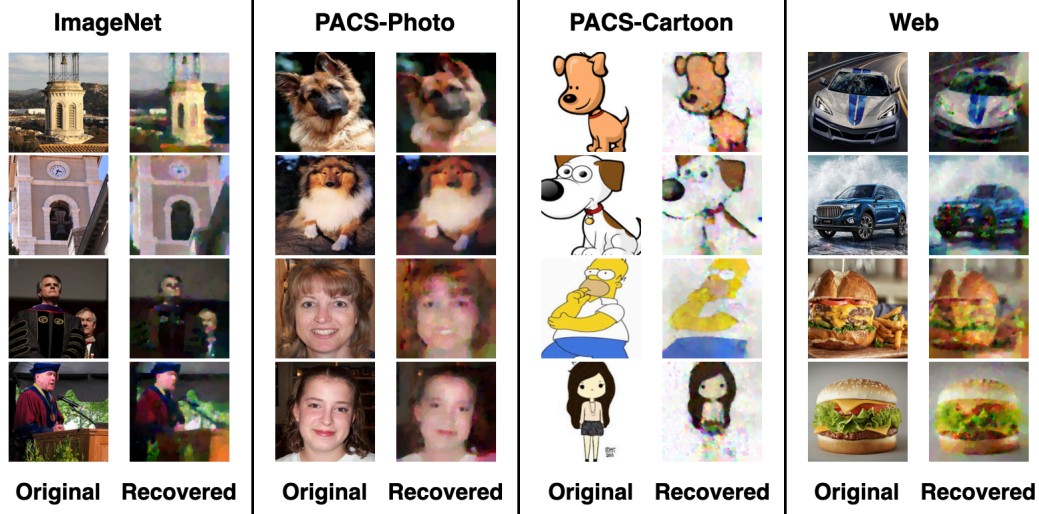

Figure 11: More reconstruction results on images from ImageNet, PACS, and Web.

## K. LIMITATIONS AND POTENTIAL DEFENSE

**Limitations.**    Following prior works (Zhu et al., 2019; Fang et al., 2023; Geiping et al., 2020), we assume an honest-but-curious server that can legitimately access per-client model updates. Our study does not consider other threat models. For instance, (1) a malicious server that actively manipulates the global model may be able to amplify the privacy-leakage risks. (2) FL systems employing MPC-based secure aggregation can preclude reliable access to per-client updates (Bonawitz et al., 2016), thereby substantially weakening our attack. We regard a systematic study of these scenarios as a promising direction for future work. In addition, as shown in Figure 8, our reconstruction quality degrades as the local batch size increases. For sufficiently large batches, our attack becomes ineffective.

**Potential defenses.**    Because our method relies on access to per-client updates, reliable multi-party computation protocols can substantially diminish its effectiveness. However, such defenses incur substantial computation and communication overhead, introducing a trade-off between utility and safety. Besides, previous studies have also discussed possible avenues to circumvent such protocols (Pasquini et al., 2022). A simpler defense is to use a large local batch size: increasing the local batch size reduces per-sample signal and tends to weaken our attack. For sufficiently large batches, effective reconstruction can no longer be feasible.

## L. LLM USAGE

We used large language models (LLMs) as general-purpose writing aids to polish wording and improve grammar.

