# OpenReview forum: "Gradient Inversion Attacks Beyond SGD"
_ICLR.cc/2026/Conference — Submitted to ICLR 2026_

### Official Review · Reviewer_CajD · 2025-10-16

**Soundness:** 2
**Presentation:** 2
**Contribution:** 3
**Rating:** 4
**Confidence:** 5

**Summary:**

This paper investigates an interesting problem in federated learning, where the authors propose the first attack approach towards FL systems with various optimizers, such as Adam. To this end, this paper designs an analytical rule for recovering labels from real optimizer updates and proposes an update-matching objective that optimizes dummy inputs to match the observed updates. Extensive experiments show that the proposed strategy exhibits great efficacy.

**Strengths:**

1. This paper discusses an important scenario beyond simply using SGD, which fills the blank in this field.
2. The authors propose two novel techniques, i.e., an analytical rule for label recovery from optimizer updates and an update-matching optimization objective, which effectively recover private images from gradients.
3. The authors provide sufficient experiments and reveal the superiority of the proposed attack.
4. The analytical method for label extraction is correct and interesting.

**Weaknesses:**

1. From my view, the major contributions of this paper lie in the novel label extraction technique, as the so-called update-matching objective is an intuitive extension of the previous loss function.
While I appreciate the derivation of this label inference technique, I do think the authors should reorganize this work to emphasize their core contributions.

2. Moreover, why don't the authors combine the proposed techniques with existing approaches? I believe the optimization objective and label inference can be effectively combined with existing gradient inversion methods, such as GIAS and GIFD. Considering the proposed mechanisms as a plug-and-play strategy for transferring existing methods to non-SGD optimization settings would better highlight your contributions.

3. The authors refer to Figures 1 and 2 in Sec. Introduction. However, there is a large gap between the text and the figure, affecting the readability of the manuscript.

**Questions:**

I'm curious about the performance of generative attacks (e.g., GIFD) combined with the proposed methods under the Adam optimizer.

---

> ### Author Response · Authors · 2025-12-03
>
> We truly appreciate the reviewer for the constructive comments.
>
> > The major contributions of this paper lie in the novel label extraction technique, as the so-called update-matching objective is an intuitive extension of the previous loss function.
>
> Thank you for your thoughtful comment. While the proposed objective function can be viewed as an extension of the previous objective function to adaptive optimizers, we would like to clarify that its effect goes beyond a straightforward extension.
>
> Under the standard SGD setup discussed in Section 4.3, our method does not collapse to the previous approach. Instead, it achieves stronger reconstruction performance, with a 3.0 dB improvement in PSNR compared with the baselines. This improvement directly stems from the objective function we introduced.
>
> > Considering the proposed mechanisms as a plug-and-play strategy for transferring existing methods to non-SGD optimization settings would better highlight your contributions. I'm curious about the performance of generative attacks (e.g., GIFD) combined with the proposed methods under the Adam optimizer.
>
> Thanks for the insightful suggestion. We report in the table below the quantitative results of using the proposed mechanisms as a plug-and-play strategy for transferring GIFD to the Adam setting. Except for switching to Adam and using images of size 224×224, we follow the original GIFD experimental setup (e.g.,  $B=1$). The same original images as those in Section 4.1 (from ImageNet) are used.
>
> Table 1. Quantitative results for GIFD combined with our strategy in the Adam Setting.
> | MSE    | PSNR    | SSIM   | LPIPS-A | LPIPS-V |
> |--------|---------|--------|---------|---------|
> | 0.0084 | 22.4748 | 0.6299 | 0.2542  | 0.3782  |
>
> As shown, combining the proposed strategy with GIFD yields high-quality reconstructions under the Adam optimizer.

---

### Official Review · Reviewer_KHeN · 2025-10-26

**Soundness:** 3
**Presentation:** 3
**Contribution:** 2
**Rating:** 4
**Confidence:** 2

**Summary:**

This paper addresses a significant gap in the field of Gradient Inversion Attacks (GIAs) by developing a method effective against Federated Learning (FL) systems that use adaptive optimizers (like Adam, RMSProp, AdaGrad), whereas prior attacks primarily worked only with the vanilla Stochastic Gradient Descent (SGD) optimizer.

**Strengths:**

1. This paper extends the threat model of Gradient Inversion Attacks to the realistic and prevalent scenario where clients use adaptive local optimizers.
2. By introducing a method to analytically recover labels from optimizer updates and a powerful update-matching objective for image reconstruction, the authors demonstrate that privacy risks in FL are more severe than previously thought, as adaptive optimizers alone do not provide sufficient protection.
3. This paper highlights a need for developing new, robust defense mechanisms tailored to secure FL systems that employ modern optimization techniques.

**Weaknesses:**

1. The attack's success is predicated on a very specific and powerful threat model that may not always hold in practice. The attack requires the attacker to know the exact initial state (e.g., $m_{t-1}$, $v_{t-1}$) of the client's optimizer at the start of the local training round. While the paper justifies this by citing FL algorithms that synchronize this state, many practical and privacy-preserving implementations do not. If a client uses a locally maintained, non-reset state unknown to the server, the entire analytical label recovery method, which is the cornerstone of the attack, would fail.
2. The attack is demonstrated on a single local update step. In real FL, clients typically perform multiple local steps before sending an update. It is unclear how the attack would perform when the observed update is an aggregate of multiple optimization steps, which would significantly obfuscate the relationship between the final update and the data from the first step.
3. All main experiments are conducted on ResNet-18. It is unclear how the attack would fare against architectures without a traditional fully-connected final layer (e.g., Transformers using a linear projection, or models with heavy use of GroupNorm) where the derived gradient constraints may not hold.
4. The attack is exclusively validated on image data. A discussion on the potential applicability to other modalities like text (where labels might be sequences) or tabular data is absent.

**Questions:**

1. How would your attack perform in a more challenging and realistic scenario where the local optimizer state is not reset and synchronized by the server at the beginning of each round, but is instead maintained privately by the client? Would the attack still be feasible without precise knowledge of $m_{t-1}$ and $v_{t-1}$?
2. The attack is demonstrated for a single local training step. What is the expected degradation in reconstruction quality if a client performs K>1 local steps, and the server only observes the aggregated parameter update? Could your method be extended to this more common FL setting?
3. The label recovery method in the "post-initial" phase relies on the observation that ambiguous gradient elements cluster in specific rows. Can you provide a theoretical intuition or more rigorous empirical analysis for why this clustering occurs? Is this a general phenomenon or something dependent on the specific dataset and model?
4. The "initial phase" label recovery critically depends on Approximation 1 (Inter-class Low Entanglement). Can you quantify the sensitivity of your attack to the validity of this approximation? How does the reconstruction quality degrade as the model moves away from this "initial" state and the approximation becomes less valid?
5. You mention secure aggregation as a defense, but do not show results. Have you tested your attack in a setting where updates from even a small number of clients (e.g., 2-10) are aggregated? Furthermore, how does your method perform against other standard defenses like the addition of differential privacy noise or aggressive gradient compression?
6. Your gradient constraints are derived for a standard fully-connected final layer. Can your label recovery method be directly applied to models with a different final layer structure, such as a Vision Transformer (ViT) which uses a single linear layer on the [CLS] token? If not, what modifications would be required?

---

> ### Author Response · Authors · 2025-12-03
>
> We truly appreciate the reviewer for the constructive comments.
>
> > The attack requires the attacker to know the exact initial state of the client's optimizer at the start of the local training round. If a client uses a locally maintained, non-reset state unknown to the server, the entire analytical label recovery method, which is the cornerstone of the attack, would fail.
>
> Thanks for your comment. The effectiveness of our method indeed relies on the assumption of accessing the initial optimizer state. Prior studies have shown that running an adaptive optimizer independently on each FL client can lead to convergence issues [1]. To address this, many existing adaptive FL algorithms unify the optimizer state across clients at the beginning of each local training round.
>
> Therefore, although the assumption may not always hold, the threat highlighted by our method remains valid, as the assumption can be satisfied in certain established adaptive FL algorithms [1,2,3]. At the same time, we regard relaxing this assumption as an important direction for future research.
>
> > The attack is demonstrated on a single local update step. Could the method be extended to the setting where a client performs K>1 local steps?
>
> Thank you for the question. Our current work focuses on the single-step setting. In the multi-step case, the proposed method cannot achieve effective reconstruction, primarily because the analytical label inference relies on mathematical properties of single-step updates and becomes unreliable under multi-step model updating.
>
> We respectively note, however, that even in this single-step scenario, no prior work has demonstrated high-quality reconstruction under modern adaptive optimizers. In addition, achieving reliable gradient inversion in multi-step settings remains a problem that has not been fully resolved even in the traditional SGD case [4]. We regard addressing this limitation as an important direction for future work.
>
> > The attack is exclusively validated on image data.
>
> Thanks for the comment. We agree that extending the analysis to other modalities, such as text or tabular data, is an important direction. We are actively exploring how the proposed mechanisms may generalize beyond the image domain.
>
> > How does the reconstruction quality degrade as the model moves away from the initial training phase?
>
> Thank you for the question. In Appendix D, we present the performance of our method at different stages of training, evaluated using the average LPIPS values. As training progresses, the LPIPS values gradually increase, indicating a decline in similarity between the reconstructed and original images.
>
> In the early phase of training, for example within the first 100 iterations, our method can produce reconstructions that closely resemble the original images, with LPIPS values below 0.6. After around 10000 iterations, the method hardly recovers meaningful information.
>
> > The paper does not show the results on secure aggregation as a defense.
>
> Thanks for the comment. We report in the table below the reconstruction performance of our method when secure aggregation is applied as a defense. In this setting, the model updates from four clients are aggregated, and the adversary only observes the aggregated update rather than each individual update.
>
> Table 1. Quantitative Reconstruction Results Under Secure Aggregation Defense
> | Defense  | MSE  $\uparrow$  | PSNR $\downarrow$   | SSIM $\downarrow$  | LPIPS-A $\uparrow$ | LPIPS-V $\uparrow$ |
> |--------|--------|---------|--------|---------|---------|
> | No Defense  | 0.0099 | 20.5985 | 0.5026 | 0.4230  | 0.5324  |
> | Secure Aggregation  | 0.1527 | 8.9056 | 0.1895 | 0.8901  | 0.8805  |
>
> As shown, secure aggregation significantly reduces the similarity between the reconstructed and real images. Compared with the no-defense case, the reconstruction quality degrades substantially, with higher MSE, lower PSNR and SSIM, and noticeably larger LPIPS values. These results demonstrate that secure aggregation can effectively defend against the proposed attack.
>
> [1] Wu, Xidong, et al. "Faster adaptive federated learning." Proceedings of the AAAI conference on artificial intelligence. Vol. 37. No. 9. 2023.
> [2] Karimireddy, Sai Praneeth, et al. "Mime: Mimicking centralized stochastic algorithms in federated learning." arXiv preprint arXiv:2008.03606 (2020).
> [3] Sun, Yan, et al. "Efficient federated learning via local adaptive amended optimizer with linear speedup." IEEE Transactions on Pattern Analysis and Machine Intelligence 45.12 (2023): 14453-14464.
> [4] Zhao, Joshua C., et al. "Loki: Large-scale data reconstruction attack against federated learning through model manipulation." 2024 IEEE Symposium on Security and Privacy (SP). IEEE, 2024.

---

### Official Review · Reviewer_1dEy · 2025-10-28

**Soundness:** 3
**Presentation:** 3
**Contribution:** 2
**Rating:** 2
**Confidence:** 5

**Summary:**

This paper presents a novel Gradient Inversion Attack (GIA) tailored for federated learning (FL) that uses adaptive optimizers (e.g., Adam), where only momentum-based updates, not raw gradients, are accessible. The method employs an update-matching objective for image reconstruction and an analytical rule to recover labels from the observed updates. Experiments on ImageNet and PACS show it significantly outperforms existing gradient-matching techniques, even achieving stronger attacks in the traditional SGD setting.

**Strengths:**

1. The paper addresses a gap in GIAs by successfully targeting FL systems that use modern adaptive optimizers like Adam. Prior GIAs largely assumed direct access to raw SGD gradients, but this work shows how to attack the more complex momentum-based updates.

2. The proposed method, which utilizes an update-matching objective and analytical label recovery, significantly outperforms previous GIAs. It achieves notably higher PSNR and SSIM scores, resulting in reconstructions that closely resemble the original private data.

3. The "update-matching" objective is robust, generalizing effectively across various adaptive optimizers (RMSProp, Adagrad, etc.). Furthermore, the new objective function provides a stronger attack, even surpassing traditional gradient-matching attacks in the standard SGD setting.

**Weaknesses:**

1. The paper's overall contribution is trivial, as the image optimization objective (Eq. 3) is an incremental change, substituting gradients with optimizer updates in an established cosine similarity loss similar to IG (Geiping et al., 2020). Moreover, the analytical label recovery relies on existing techniques, drawing foundational principles from GIAs (Eq. 6) and assumptions (Approximation 1 and Assumption 1) previously proposed in iLRG (Ma et al., 2023) and GradientInv (Yin et al., 2021), respectively. Consequently, the main technical novelty is confined to the specific derivation that infers the final-layer gradient information exclusively from the Adam-like optimizer's update parameters.

2. The paper overstates its first challenge (Objective Function) since the optimizer update $\mathcal{U}(\nabla\theta, s)$ shares the same dimensionality as the raw gradient $\nabla\theta$ and model parameters $\theta$. Although it is not the exact gradient $\nabla\theta$, $\mathcal{U}(\nabla\theta, s)$ differs only by a further step of computation. Hence, it is very straightforward for gradient matching.

3. The attack is significantly constrained as its high efficacy is primarily limited to the initial training phase, with performance severely degrading thereafter. Furthermore, the analytical label recovery for this critical early phase relies on highly idealized constraints, such as the Inter-class Low Entanglement Approximation and Non-negative Activation Assumption.

**Questions:**

1. Given that label recovery is core to the attack, what is the quantitative accuracy of the inferred labels across various model architectures and datasets used in your experiments?

2. Assumption 1 only holds for ReLU and Sigmoid. How can the analytical label recovery method be adapted to accommodate other activation functions, such as Tanh or GeLU?

3. If clients perform multiple local training updates before sending the final aggregation, how does this process of local averaging affect the overall effectiveness of the proposed attack?

4. The theory primarily derives from the Adam update rule (Eq. 4). How does a differing mechanism in other adaptive optimizers affect the generalizability and theoretical soundness of the label recovery process?

---

> ### Author Response · Authors · 2025-12-03
>
> We truly appreciate the reviewer for the constructive comments.
>
> > The paper's overall contribution is trivial, as the image optimization objective (Eq. 3) is an incremental change, substituting gradients with optimizer updates in an established cosine similarity loss similar to IG.
>
> Thanks for your comment. While the proposed objective function can be viewed as an extension of the previous objective function to adaptive optimizers, we would like to clarify that its effect goes beyond a straightforward extension.
>
> Under the standard SGD setup discussed in Section 4.3, our method does not collapse to the previous approach. Instead, it achieves stronger reconstruction performance, with a 3.0 dB improvement in PSNR compared with the baselines. This gain arises from the objective function we introduced.
>
> > The analytical label recovery for this critical early phase relies on highly idealized constraints, such as the Inter-class Low Entanglement Approximation and Non-negative Activation Assumption.
>
> Thank you for the comment. We agree that our method builds upon the Inter-class Low Entanglement Approximation and the Non-negative Activation Assumption. We respectfully clarify, however, that these conditions are not purely idealized.
>
> Prior studies have shown that they can indeed be approximately satisfied in practical settings [1,2], though they may not hold in all cases. Therefore, our method is not effective only under idealized constraints but also reflects a practical privacy threat in certain realistic scenarios.
>
> > If clients perform multiple local training updates before sending the final aggregation, how does this process of local averaging affect the overall effectiveness of the proposed attack?
>
> Thank you for the question. Our current work focuses on the single-step setting. In the multi-step case, the proposed method cannot achieve effective reconstruction, primarily because the analytical label inference relies on mathematical properties of single-step updates and becomes unreliable under multi-step model updating.
>
> We respectively note, however, that even in this single-step scenario, no prior work has demonstrated high-quality reconstruction under modern adaptive optimizers. In addition, achieving reliable gradient inversion in multi-step settings remains a problem that has not been fully resolved even in the traditional SGD case [3]. We regard addressing this limitation as an important direction for future work.
>
> > The theory primarily derives from the Adam update rule (Eq. 4). How does a differing mechanism in other adaptive optimizers affect the generalizability and theoretical soundness of the label recovery process?
>
> Thank you for the question. Since Adam is one of the most widely used adaptive optimizers, we use it to present the analysis. However, the theoretical derivation is not specific to Adam. For other commonly used adaptive optimizers such as Adagrad and RMSProp, Eq. 4 can be replaced with the corresponding update rule, and the subsequent derivation remains valid, as the required mathematical properties are preserved.
>
> [1] Ma, Kailang, et al. "Instance-wise batch label restoration via gradients in federated learning." The Eleventh International Conference on Learning Representations. 2023.
> [2] Yin, Hongxu, et al. "See through gradients: Image batch recovery via gradinversion." Proceedings of the IEEE/CVF conference on computer vision and pattern recognition. 2021.
> [3] Zhao, Joshua C., et al. "Loki: Large-scale data reconstruction attack against federated learning through model manipulation." 2024 IEEE Symposium on Security and Privacy (SP). IEEE, 2024.

---

### Official Review · Reviewer_5Jp8 · 2025-11-01

**Soundness:** 3
**Presentation:** 3
**Contribution:** 3
**Rating:** 6
**Confidence:** 3

**Summary:**

The paper broadens Gradient Inversion Attacks to the realistic case where the adversary observes parameter updates rather than raw gradients. It provides an analytic label-recovery step from updates plus a general update-matching loss built from the optimizer rule, achieving sizable reconstruction gains and even outperforming gradient-matching in standard SGD.

**Strengths:**

1. A clean reframing of the threat model covering adaptive optimizers used in practice.

2. Strong empirical lift across optimizers/datasets, including large-scale cases.

**Weaknesses:**

1. Reliance on optimizer state knowledge. The approach assumes access to optimizer state each round like moments and variance. In many federated settings, such state may be hidden. Even small mismatches can degrade label algebra or update alignment, potentially reducing attack reliability.

2. Uncertain robustness under defenses. Schemes like secure aggregation, compression, randomized quantization, or DP noise could distort updates and optimizer states. The paper shows strong results without such defenses, but doesn’t quantify how much distortion is needed to break the analytic step.

**Questions:**

See weaknesses.

---

> ### Author Response · Authors · 2025-12-03
>
> We truly appreciate the reviewer for the constructive comments.
>
> > Reliance on optimizer state knowledge. The approach assumes access to optimizer state each round, like moments and variance. In many federated settings, such state may be hidden.
>
> Thanks for your comment. Prior studies have shown that running an adaptive optimizer independently on each FL client can lead to convergence issues [1]. To address this, many existing adaptive FL algorithms unify the optimizer state across clients at the beginning of each local training round.
>
> Therefore, although the assumption may not always hold, the threat highlighted by our method remains valid, as the assumption can be satisfied in certain established adaptive FL algorithms [1,2,3]. At the same time, we regard relaxing this assumption as an important direction for future research.
>
> > Uncertain robustness under defenses. The paper shows strong results without defense, but doesn’t quantify how much distortion is needed to break the analytic step.
>
> Thanks for the comment. We report in the table below the reconstruction performance of our method when secure aggregation is applied as a defense. In this setting, the model updates from four clients are aggregated, and the adversary only observes the aggregated update rather than each individual update.
>
> Table 1. Quantitative Reconstruction Results Under Secure Aggregation Defense
> | Defense  | MSE  $\uparrow$  | PSNR $\downarrow$   | SSIM $\downarrow$  | LPIPS-A $\uparrow$ | LPIPS-V $\uparrow$ |
> |--------|--------|---------|--------|---------|---------|
> | No Defense  | 0.0099 | 20.5985 | 0.5026 | 0.4230  | 0.5324  |
> | Secure Aggregation  | 0.1527 | 8.9056 | 0.1895 | 0.8901  | 0.8805  |
>
> As shown, secure aggregation significantly reduces the similarity between the reconstructed and real images. Compared with the no-defense case, the reconstruction quality degrades substantially, with higher MSE, lower PSNR and SSIM, and noticeably larger LPIPS values. These results demonstrate that secure aggregation can effectively defend against the proposed attack.
>
>
> [1] Wu, Xidong, et al. "Faster adaptive federated learning." Proceedings of the AAAI conference on artificial intelligence. Vol. 37. No. 9. 2023.
> [2] Karimireddy, Sai Praneeth, et al. "Mime: Mimicking centralized stochastic algorithms in federated learning." arXiv preprint arXiv:2008.03606 (2020).
> [3] Sun, Yan, et al. "Efficient federated learning via local adaptive amended optimizer with linear speedup." IEEE Transactions on Pattern Analysis and Machine Intelligence 45.12 (2023): 14453-14464.

---

### Meta-Review · Area_Chair_nT1k · 2025-12-08

**Summary:**

The paper proposes a gradient inversion attack tailored for Federated Learning systems using adaptive optimizers (e.g., Adam), moving beyond the standard SGD setting. The authors introduce an analytical label recovery method and an update-matching objective.

While the reviewers acknowledged the motivation of extending attacks to adaptive optimizers, the consensus is that the proposed threat model is restrictive and fragile. First, the method's fundamental limitation is the single-step local updates, which contradict standard Federated Learning practices like FedAvg, where clients perform multiple local steps, and its demonstrated vulnerability under basic defenses. Therefore, I recommend this paper for rejection.

**Reviewer Concerns:**

Regarding the reviewer concerns, the authors successfully addressed specific technical requests, such as integrating their method with GIFD, as requested by Reviewer CajD, and citing literature to justify the synchronized optimizer state assumption raised by Reviewer 5Jp8.

However, the most critical concerns regarding the practicality and robustness of the attack remain outstanding. Specifically, Reviewers 1dEy and KHeN identified that the attack is limited to single-step local updates, and the authors admitted in the rebuttal that the method fails in the multi-step settings common to practical FL, which severely limits the work's applicability. Furthermore, while the authors provided new data on Secure Aggregation in response to Reviewers 5Jp8 and KHeN, these results confirmed that the attack is easily mitigated by standard defenses, thus validating the concern that the proposed threat is fragile and lacks practical impact.

**Reviewer Scores:**

Reviewers 5Jp8, 1dEy, and KHeN may maintain their scores because of the concerns of robustness against defense methods and single-step limitations. Reviewer CajD might slightly raise their score (e.g., from 4 to 6) due to the successful inclusion of the requested GIFD experiments.

---

### Decision · Program_Chairs · 2026-01-26

Reject